# The Safety and Feasibility of Lower Body Positive Pressure Treadmill Training in Individuals with Chronic Stroke: An Exploratory Study

**DOI:** 10.3390/brainsci13020166

**Published:** 2023-01-18

**Authors:** Sattam M. Almutairi, Moodhi M. Alfouzan, Taghreed S. Almutairi, Hatem A. Alkaabi, Misoon T. AlMulaifi, Marzouq K. Almutairi, Faisal K. Alhuthaifi, Chad Swank

**Affiliations:** 1Department of Physical Therapy, College of Medical Rehabilitation Science, Qassim University, Buraydah 52571, Saudi Arabia; 2Rehabilitation Department, Qassim University Medical City, Buraydah 52571, Saudi Arabia; 3Baylor Scott & White Research Institute, Dallas, TX 75246, USA

**Keywords:** rehabilitation, gait disorder, assistive technology, locomotion, stroke

## Abstract

Background: Lower body positive pressure (LBPP) may provide a novel intervention for gait training in neurological conditions. Nonetheless, studies investigating the safety and feasibility of LBPP in patients with stroke are insufficient. Objectives: The purpose of this study was to evaluate the safety and feasibility of LBPP as a rehabilitation intervention for individuals with chronic stroke. Methods: Individuals with chronic stroke were recruited from the community to participate in LBPP gait training three times a week for six weeks. The LBPP’s safety and feasibility were documented throughout the study and at the end of six weeks. Safety and feasibility referred to the incidence of adverse events, complications, the participant and therapist satisfaction questionnaire, and the device limitation including but not limited to technical issues and physical constraints. In addition, blood pressure, pulse rate, and oxygen saturation were taken pre- and post-session. Dependent *t*-tests were used to analyze the difference between assessments. A Wilcoxon test was used to assess the ordinal data (Trial registration number NCT04767334). Results: Nine individuals (one female, eight males) aged 57 ± 15.4 years were enrolled. All participants completed the intervention without adverse events. All participants reported positive scores from 4 (very satisfying) to 5 (extremely satisfying) in the safety and feasibility questionnaire. No significant differences were observed in blood pressure and oxygen saturation during the intervention sessions. However, significant increases were observed in heart rate from 82.6 ± 9.1 beats/min (pre-session) to 88.1 ± 6.8 beats/min (post-session) (*p* = 0.027). Conclusions: LBPP is a safe and feasible rehabilitation tool to use with individuals with chronic stroke.

## 1. Introduction

Stroke is a neurological disease characterized by chronic major disability worldwide [1], which causes permanent changes in an individual’s life [2]. There are 101 million people living with stroke around the world [1]. In Saudi Arabia, stroke is the second-largest cause of disability and death [3,4]. Many individuals with stroke have mobility deficits such as slower walking speed and decreased postural control [5], which contribute to an increased risk of falls, disruption of activities of daily living, and reduced quality of life [6,7]. Recovery of independent walking is a fundamental goal for post stroke survivors and clinicians in rehabilitation practice [8].

Body-weight supported treadmill training (BWSTT) is recommended for gait training in contemporary rehabilitation practice [9] and has been shown to increase walking speed and walking endurance [9]. The unloading of body weight allows for lower limb movement for gait training and reduces forces across the knee and ankle joint [10]. However, BWSTT is beset by challenges of additional set-up and potentially more than two clinicians are needed for a session [11]. Additionally, when compared to other types of gait training, BWSTT was not shown to be more effective in improving the walking ability [9]. In contrast, a lower body positive pressure (LBPP) system capitalizes on the benefits of unloading body weight for gait training and minimizes the challenges of BWSTT. LBPP is a treadmill installed in an enclosed inflated bag to reduce up to 80% of the body weight during gait training with minimal gait kinematic alteration [12,13]. Furthermore, concurrent visual feedback is provided to the patient during the task-specific gait training through a video camera and screen attached to contemporary LBPP systems [14].

The physiological response of healthy individuals to LBPP has been extensively evaluated [12,15,16,17]. The findings from these studies indicate that LBPP is safe and does not adversely impact cardiovascular responses. In particular, blood pressure, brain oxygenation, blood flow velocity through the middle cerebral artery, and head skin microvascular blood flow did not change significantly when using LBPP [12]. Consequently, LBPP has been utilized as a rehabilitation tool to improve dynamic balance and gait speed for people with postoperative orthopedic conditions [18,19], Parkinson’s disease [20], patients with stroke [13,21,22,23,24], and children with cerebral palsy [25].

Nevertheless, studies on the safety and feasibility of LBPP in patients with stroke are lacking. Few studies have assessed the effectiveness of LBPP in individuals with chronic stroke [13,21,22,23,24]. However, the safety and feasibility of using LBPP in individuals with chronic stroke have not been formally assessed. Patients with stroke have high rates of cardiovascular system disorders such as recurrent cerebrovascular accidents, coronary artery disease, and hypertension [26]. Unexpected changes to the cardiovascular system might result in a serious medical condition. LBPP uses an air compressor to increase the pressure inside the chamber above the external ambient pressure [12]. This pressure causes an axial buoyant force, allowing patients to ambulate on the treadmill under simulated fractional gravity [12]. Thus, LBPP needs to be safe in order to be considered as an effective rehabilitation tool in the stroke population. To the best of our knowledge, no one has formally investigated the safety and feasibility of using LBPP in people with chronic stroke. As a result, the purpose of this study was to evaluate the safety and feasibility of LBPP as a rehabilitation tool for individuals with chronic stroke. We hypothesize that LBPP is safe and feasible for gait training for individuals with chronic stroke.

## 2. Methods

### 2.1. Study Design

This was a an exploratory study with a single group receiving the same intervention and measures. The study protocol was approved by the Institutional Review Board at Qassim University (reference number 20-05-04) and registered at ClinicalTrials.gov with ID number NCT04767334.

### 2.2. Participants

A convenience sample of individuals aged between 18 and 70 years with chronic stroke was recruited through local advertising and word-of-mouth by co-investigators in the Qassim region. All participants signed an informed consent form before testing or data collection. The principal investigator (PI) explained the research project and its risks and benefits to participants upon enrollment. Next, a medical history and demographic questionnaire were completed. Participants were screened by an experienced physical therapist for eligibility to participate in this study. Inclusion criteria were: (1) hemiparesis caused by brain stroke; (2) a minimum of 6 months since the stroke; (3) able to walk a minimum of 10 m with or without assistive device; (4), have no other neurological or orthopedic issues that affect ambulation; (5) functional ambulation ≥3 in the functional ambulation category [27]; (6) no preexisting respiratory or cardiovascular conditions that interfere with protocol; (7) the ability to comprehend basic instructions; and (8) the ability to maintain postural control. Exclusion criteria were (1) recurrent stroke; (2) lower limb spasticity greater than 3 on the modified Ashworth scale; and (3) ataxia or tremor of the lower limb.

Demographic and clinical variables: Participant characteristics, self-reported comorbidities such as diabetes and hypertension, self-reported medications, and self-reported history of falls in the previous 12 months, were obtained at the baseline.

Safety and feasibility: The safety and feasibility of the LBPP were evaluated by reporting adverse events such as falls or fainting, complications, attrition rate, the participants’ and therapists’ satisfaction questionnaire, and the device limitation. The participants and therapist satisfaction questionnaire items (safety, convenience, size, personal acceptance, and ease of operation) were assessed using a numerical rating scale rating from 0 (extremely dissatisfied) to 5 (extremely satisfied) [28] (Appendix A).

In addition, we assessed and reported vital signs including heart rate, blood pressure, and oxygen saturation at the beginning of LBPP gait training and at the end of each session. Heart rate, blood pressure, and oxygen saturation were measured utilizing the Welch Allyn Connex Spot Monitor (CSM) (Welch Allyn, Skaneateles Falls, NY, USA) with a manual cuff applied to the left arm. The participants and therapists were asked to rate their level of satisfaction for each satisfaction questionnaire item at the end of the intervention protocol. The time and number of sessions were documented for each participant.

Functional Ambulation Category (FAC): This is a clinical gait assessment that categorizes walking ability into six levels based on the amount of physical support required, regardless of assistive device use. It ranges from 0 to 5, where 0 indicates that the patient cannot walk (non-functional ambulation) and 5 indicates that the patient can walk without physical support on any surface (independent ambulation) [27]. The FAC was measured at the baseline and after the intervention protocol.

### 2.3. Procedure

After providing informed consent, participants completed a demographic, medical history, and surgical history form. Additionally, they were evaluated for eligibility.

Each participant started by warming up on a standard cycle ergometer for 5–10 min, followed by manual therapy, therapeutic exercises, and LBPP gait training. Manual and therapeutic exercise consisted of passive and active ROM, joint mobilization, passive and active stretching, manual resistance exercise, postural control and balance, and upper extremity control. To use the LBPP, participants wore neoprene shorts with a kayak-type skirt attached at the level of the waist (Figure 1). Then, the participants stepped into the LBPP chamber, usually with assistance from two physical therapists to assure a safe transfer. Once the participant was inside the chamber, the neoprene skirt comfortably sealed over the lip at the top of the chamber. Once the participant felt comfortable standing inside the chamber, the physical therapist gave them the instructions to be ready for gait training.

For gait training, all participants walked in LBPP one session a day (for up to approximately 40 min), three times a week, for a total of six weeks. During the gait training, the camera provided video feedback of each foot strike. For each session, before and after the LBPP gait training, vital signs including heart rate, blood pressure, and oxygen saturation were taken. In session one, the LBPP chamber was set to unload 50% of the participant’s body weight. The percentage of the unload decreased gradually by approximately 2% per session in the following sessions. Based on the participant’s capacity, the physical therapist’s support and treadmill speed were evaluated and adjusted accordingly. During the gait training, the participant could stop whenever they needed to.

### 2.4. Statistical Analyses

The SPSS (Version 28.0) software program was used for all statistical analyses. Descriptive statistics including the means (with standard deviations) and median (with interquartile range (IQR)) were calculated for the demographic variables and main outcome measures. The safety and feasibility of the LBPP gait training were assessed by the reporting of adverse events, complications, attrition rate, participant and therapist satisfaction questionnaires, and the device limitations. In addition, we reported vital signs such as heart rate, blood pressure, and oxygen saturation before and after each LBPP gait training session. Comparisons between the pre- and post-sessions for vital signs were conducted using a dependent *t*-test to determine whether there were any significant differences between the two time intervals. Ordinary data such as FAC were assessed using the Wilcoxon test. Differences were considered statistically significant when *p* < 0.05.

## 3. Results

Thirteen participants were recruited to enroll in the study. Four participants dropped out due to their inability to complete the required sessions. One participant (*n* = 1) completed 16 sessions before withdrawing due to cold weather and transportation issues; another participant (*n* = 1) completed six sessions before withdrawing due to family circumstances; and two other participants (*n* = 2) completed 12 sessions before withdrawing due to orthopedic problems (fall; knee pain). The fall and knee pain were not related to the study intervention. Therefore, nine participants completed the 18 sessions and were included in the analysis (flow diagram in Figure 2).

Participants (one female, eight males) aged 57 ± 15.4 year with chronic stroke for 4.8 ± 3.9 years. The height of the participants was 175 ± 16.4 cm, and their weight was 80 ± 11.6 kg. The demographic characteristics of the participants are listed in Table 1. Participants walked without physical assistance (FAC, range = 3 to 5).

All participants completed the rehabilitation protocol without adverse events. The participants and therapists reported positive satisfaction scores ranging from 4 (very satisfying) to 5 (extremely satisfying) in the therapist and participant satisfaction questionnaire. Two participants reported that the camera feedback was helpful for gait and posture correction; one participant complained of a noisy sound from the air compressor and the device. The LBPP treadmill’s speed ranged from 0.2 to 2.5 km/h, with an average of 0.83 km/h. The single session during gait training in LBPP lasted an average of 37.95 min. The body weight loading increased by approximately 34% over the treatment period, from 50% to 84% of the body weight.

There was a statistically significant difference in FAC before (median ± IQR; 4 ± 2) and after (median ± IQR; 5 ± 1) intervention (*p* = 0.034). Furthermore, the blood pressure, pulse rate, and oxygen saturation were documented pre- and post-session. Statistical analysis showed no significant differences in blood pressure or oxygen saturation (*p* < 0.05). The average blood pressure-systolic remained unchanged from the pre-session with a value of 143 ± 23 mmHg to post-session 141 ± 26 mmHg (*p* = 0.46). Similarly, the average blood pressure-diastolic remained unchanged from the pre-session with a value of 79 ± 2.9 mmHg to the post-session 78 ± 4.3 mmHg (*p* = 0.43). The average oxygen saturation pre-session value was 96% ± 1 compared to the post-session value of 95.2 ± 1 (*p* = 0.065). However, there was a significant increase in the pulse rate from 82.6 ± 9.1 beats/min (pre-session) to 88.1 ± 6.8 beats/min (post-session) (*p* = 0.027). Table 2 provides the details of the LBPP session.

## 4. Discussion

Except for one negative comment, the study reported no adverse events related to the LBPP device during the intervention. Despite this feedback, which regarded the noisy sound from the device, the participant’s score on the feasibility questionnaire was satisfied with the device. These findings are also consistent with a previous study, which reported no adverse events or any side effects of using LBPP with chronic stroke [13]. This study confirmed our hypothesis that LBPP is safe and feasible to use with chronic stroke.

In addition to the absence of adverse events, the participant and therapist satisfaction questionnaires were highly scored in terms of satisfaction. Furthermore, two positive feedbacks indicated that the camera feedback was helpful during gait training for gait correction, which might produce a more symmetric gait pattern. According to the literature, augmented visual feedback is a useful tool in rehabilitation practice [29] as it enhances movement performance and cortical neuroplasticity [29].

The attrition rate in this study was quietly high (31%). However, in the literature, attrition rates of 30–70% are often reported, especially when the study design is a longitudinal study over a period of time [30]. In this study, we were able to contact the dropped-out participants to find out the reason for the drop-out and to fill out the participant satisfaction questionnaire. All reasons for dropping out of the study were not related to the study protocol or the device. Nevertheless, we asked all of the participants including the dropped-out participants and therapists regarding the degree of the device’s safety, their physical and psychological well-being, the convenience of the device’s size, their motivation to use the device, and the simplicity of using the device. All participants including those who dropped-out and the therapists gave answers ranging from 4 (satisfied) to 5 (very satisfied).

This study monitored the cardiovascular responses pre and post each session. Heart rate and blood pressure are common cardiovascular response measures used to assess safe exercise intensity [15]. In this study, no significant changes were observed in the blood pressure level and oxygen saturation. Similarly, LBPP did not affect blood pressure, brain oxygenation, middle cerebral artery blood flow velocity, or head skin microvascular blood flow in healthy persons [12]. The average pulse rate, however, was increased from 82.6 ± 9.1 bpm pre-session to 88.1 ± 6.8 bpm post-session, which is a normal response due to the walking exercise intensity [15,31]. Likewise, Cutuk et al. (2005) reported that LBPP significantly increased the heart rate from 83 ± 3 bpm at rest to 99 ± 4 bpm while walking at 3.0 mph (1.34 m/s) in healthy individuals. Previous studies that demonstrated that the training on LBPP maintained the heart rate, blood pressure, and oxygen saturation on a safe level support the findings of this study [12,32].

The drawback of LBPP technology is that the therapist cannot provide manual cues to the patient’s lower extremity during gait training. Additionally, LBPP is not provided with a harness to support patients who have weaker lower extremity muscles and cannot stand in the device. Moreover, the transfer getting in and out of the device usually requires two therapists to assure the safety of patients with hemiparesis. This current study confirmed that stroke survivors can be trained on LBPP. This is, to the best of our knowledge, the first study that formally investigated the safety and feasibility of LBPP in chronic stroke. LBPP provides the patient with intense, repetitive gait training over an extended period that leads to task-specific rehabilitation. This study suggests that using LBPP might be a good decision in rehabilitation practice. It can provide the patient with higher doses of training with minimal assistance from a single therapist, which is usually sufficient. However, further studies with a larger sample size in the stroke population are needed to confirm our findings.

There were a few limitations to the current study including the relatively small sample size, which would have made it difficult to generalize the findings. Additionally, our small group was not homogeneous in disease duration or severity, which likely influenced our findings. Moreover, because our participants were, on average, community ambulators, our sample was probably not representative of all stroke patients. The LBPP requires individuals to be ambulatory; therefore, only individuals with stroke who are able to stand and walk can be trained. Further explorations of the effects of LBPP on mobility and balance function are warranted. Moreover, all participants were in a chronic stage of stroke, so further research should examine the cardiovascular and metabolic response in individuals with subacute stroke.

## 5. Conclusions

In summary, the current study showed that LBPP is safe and feasible to use with individuals with chronic stroke. Cardiovascular responses were stable and within the normal limit during pre- and post-walking exercise on the LBPP. Further investigation of the LBPP should be evaluated in a variety of neurological conditions such as spinal cord injury and multiple sclerosis.

## Figures and Tables

**Figure 1 brainsci-13-00166-f001:**
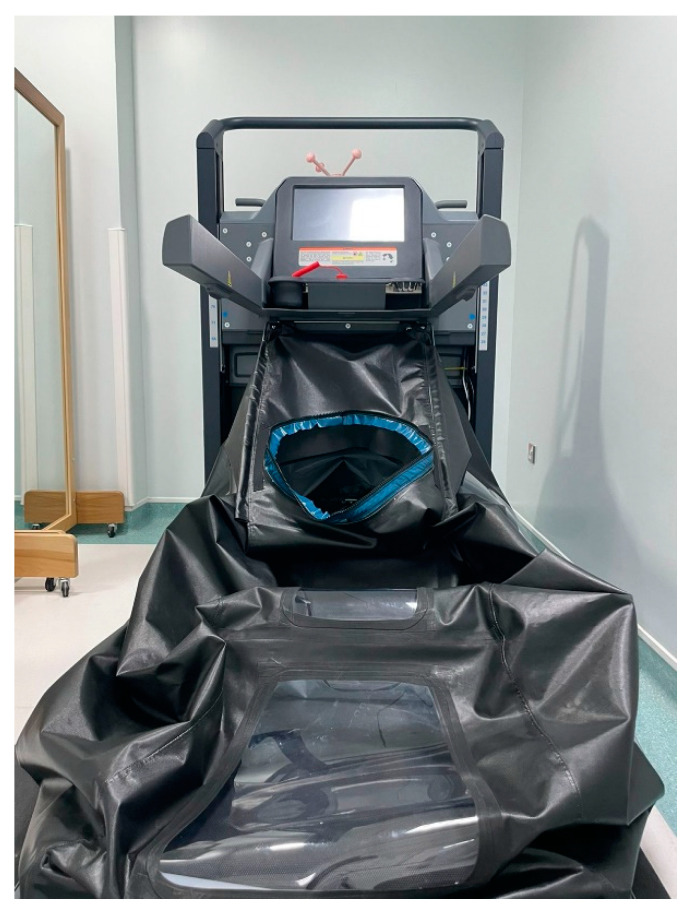
Lower Body Positive Pressure Treadmill (AlterG Anti-Gravity Treadmill).

**Figure 2 brainsci-13-00166-f002:**
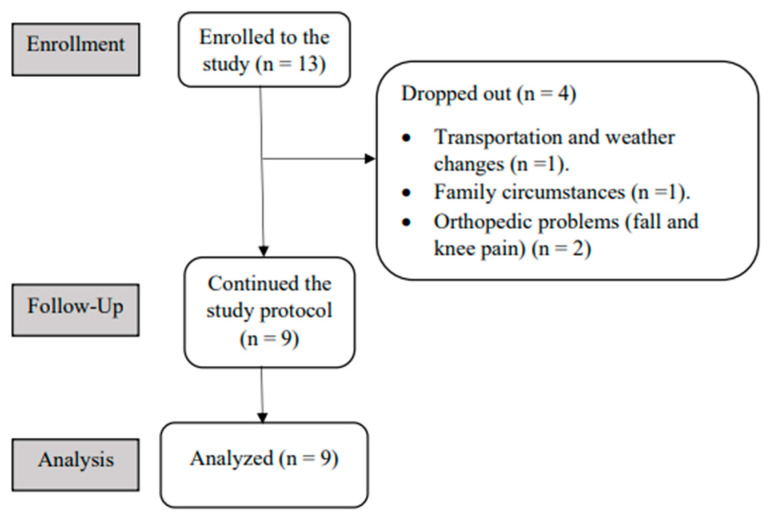
The flow of the study diagram.

**Table 1 brainsci-13-00166-t001:** The demographic characteristics of the participants (*n* = 9).

Participant Number	Age (Year)	Weight (kg)	Height (cm)	TSS (Years)	Gender	Involved Side	Fall History (*n*)	Chronic Conditions	Marital Status	AFO	Assistive Device
1	70	73.20	161	0.5	M	Right	No	DM + HTN + DYS	Married	No	Cane
2	67	86.60	155	8	M	Left	Yes (5)	DYS	Married	No	Cane
3	60	70.80	163	2	M	Right	No	DM	Married	No	Cane
4	28	64.90	175	7	M	Left	Yes (4)	No	Single	No	No
5	33	92	162	6	F	Left	No	No	Married	Yes	No
6	63	86	167	3	M	Right	Yes (2)	DM + HTN	Married	No	Cane
7	66	99	168	1.5	M	Right	No	DM + HTN + DYS	Married	No	Cane
8	64	68	158	3	M	Left	No	DM + DYS	Married	Yes	Cane
9	64	81	175	13	M	Right	No	DM + HTN + DYS	Married	No	Cane
T: (m ± SD)	57 ± 15.4	80 ± 11.6	175 ± 16.4	4.8 ± 3.9	-	-	-	-	-	-	-

Note: T = total, m = mean, SD = standard deviation, kg = kilogram, cm = centimeter, TSS = time since stroke, M = male, F = female, *n* = number of fall, DM = diabetes mellitus, HTN = hypertension, DYS = dyslipidemia.

**Table 2 brainsci-13-00166-t002:** The lower body positive pressure treadmillsession details and cardiovascular responses (*n* = 9).

Variables	(m ± SD)	*p*-Value
Speed (km/h)		
Average	0.83 ± 0.05	-
Duration (minutes)		
Average	37.59 ± 2.03	-
Unloading (%)		
Average	64.9 ± 9.5	-
Blood Pressure (mmHg)		
Pre-LBPP	(143/79) ± (23/2.9)	p (Sys) = 0.46
Post-LBPP	(141/78) ± (26/4.3)	p (Dia) = 0.43
Pulse Rate (bpm)		
Pre-LBPP	82.6 ± 9.1	*p* = 0.027 *
Post-LBPP	88.1 ± 6.8	
Oxygen Saturation (%)		
Pre-LBPP	96% ± 1	*p* = 0.065
Post-LBPP	95.2% ± 1	

Note: km/h = kilometer per hour, mmHg = millimeters of mercury, bpm = beats per minutes, min = minutes, m = mean, SD = standard deviation, Sys = systolic, Dia = diastolic, * = significant at (*p* < 0.05).

## Data Availability

Dataset analysis will be available from the corresponding author upon reasonable request after publication of the trial findings.

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
