# Peer review of "The Safety and Feasibility of Lower Body Positive Pressure Treadmill Training in Individuals with Chronic Stroke: An Exploratory Study"

_brainsci, 2023, doi:10.3390/brainsci13020166_

Round 1
Reviewer 1 Report
The author evaluates the application of Lower body positive pressure (LBPP) in chronic stroke survivors. The main measures are differences in pulse rate and the questionnaire given to both the clients and therapists.
Suggestions:
1) I am wondering if there is any break during LBPP session? What determines differences in exercise duration for each subject? Did the subjects experience an after-effect after LBPP (sensorimotor system adapts to the environment condition, e.g. walking in a spaceflight).
2) Did the authors perform verbal (qualitative) interview? Any verbal description or feedback on the LBPP method from both the clients and therapists will be beneficial to be reported, as opposed to a Likert Scale interview. If not, this can be a limitation.
3) Section 3 - 5 are parts of the Methods section and they should be as an individual sub-heading instead.
4) Since the main study is rather thin, can the authors compare the main contribution of their study in the Discussion? For example: how does LBPP affect the blood flow to the brain? Has any prior study found improvement in gait measures? This is so that the main contribution is not about safety/feasibility of LBPP.
Minor:
1) Is the affiliation for each author's name missing?
Author Response
Response to Reviewer 1 Comments
Thank you for reviewing our manuscript the safety and feasibility of lower body positive pressure treadmill training in individuals with chronic stroke. The authors appreciate the time and effort devoted to revising our manuscript. The responses to your comments and recommendations are provided below.
Point 1: I am wondering if there is any break during LBPP session? What determines differences in exercise duration for each subject? Did the subjects experience an after-effect after LBPP (sensorimotor system adapts to the environment condition, e.g. walking in a spaceflight).
Response 1: There was no break between sessions of the LBPP. Participants were asked to walk for 40 minutes. However, a few participants left the session earlier than anticipated (40 minutes). The duration of an average session has been reported. The authors did not investigate this explicitly, but we believe the participant's fitness plays a role in determining the session length. Unfortunately, we did not examine the immediate effects of LBPP, but it would be a highly interesting question for future research using advanced labs such as Vicon motion analysis and EMG.
Point 2: Did the authors perform verbal (qualitative) interview? Any verbal description or feedback on the LBPP method from both the clients and therapists will be beneficial to be reported, as opposed to a Likert Scale interview. If not, this can be a limitation.
Response 2: Unfortunately, the authors did not formally and qualitatively evaluate the participants' and therapists' feedback. However, when using the satisfaction questionnaire, we added a note about their comments. In the manuscript, we reported about two participant comments on the camera and noise.
Point 3: Section 3 - 5 are parts of the Methods section and they should be as an individual sub-heading instead.
Response 3: Fixed.
Point 4: Since the main study is rather thin, can the authors compare the main contribution of their study in the Discussion? For example: how does LBPP affect the blood flow to the brain? Has any prior study found improvement in gait measures? This is so that the main contribution is not about safety/feasibility of LBPP.
Response 4: A new sentence has been added to the manuscript (Lines 52–54). There were few studies that investigated gait and balance in stroke patients using LBPP. In the introduction, we emphasize the contribution of these studies (lines 61–64). However, since the focus of our study was to assess the safety and feasibility, the authors did not discuss the effect of LBPP on other physical functions. In addition, we suggested future research to investigate the impact of LBPP on all mobility and balance functions (Lines 77–78).
Minor:
Point 1: Is the affiliation for each author's name missing?
Response 1: Every affiliation has written.
Reviewer 2 Report
Dear Editor,
I reviewed the manuscript detailed below.
‘The Safety and Feasibility of Lower Body Positive Pressure Treadmill Training in Individuals with Chronic Stroke’
The authors investigated in 9 post-stroke patients the Lower Body Positive Pressure (LBPP) rehabilitation treatment. They focused in feasibility and safety of the procedure. The question was answered: the procedure is safe and feasible, but the most interesting question is it useful. Patient’s satisfaction increased from very satisfying to extremely satisfying. However, the paper is not that valuable and captivating, but results like these can be reported as well.
I have several concerns regarding the manuscript, so I would suggest some, which probably will improve it:
1. In the abstract I will explain what means device limitation.
2. In my opinion for the statistical analysis using median and interquartile instead mean and SD would be more appropriate, due to the small number of participants. I an normal distribution there? I wouldn’t make the results more fancy, but correct. Also reconsider the testing methods. T-test?
3. As far the manuscript addresses in particular neurologists, I would avoid propaedeutic information, such as ‘stroke is characterized by chronic disability…… .
4. On the other side, the first part of the discussion is perfect!!!
5. The rest of the discussion is comprehensively elaborated. The authors my consider some cutting (optional).
Author Response
Response to Reviewer 2 Comments
Thank you for reviewing our manuscript on the safety and feasibility of lower body positive pressure treadmill training in individuals with chronic stroke. The authors appreciate the time and effort devoted to revising our manuscript. The responses to your comments and recommendations are provided below.
Point 1: In the abstract I will explain what means device limitation.
Response 1: Accordingly, amendments have been made (line 22).
Point 2: In my opinion for the statistical analysis using median and interquartile instead mean and SD would be more appropriate, due to the small number of participants. I an normal distribution there? I wouldn’t make the results more fancy, but correct. Also reconsider the testing methods. T-test?
Response 2: For ordinal data (FAC), we used the median and interquartile range. However, the vast majority of our data is expressed as a percentage and as continuous data. Thus, the authors decided to present them with a mean and standard deviation. The small sample size has been recognized as a limitation.
Point 3: As far the manuscript addresses in particular neurologists, I would avoid propaedeutic information, such as ‘stroke is characterized by chronic disability…… .
Response 3: The authors wrote a few sentences about stroke patients' characteristics to strengthen the introductory paragraph.
Point 4:On the other side, the first part of the discussion is perfect!!!
Response 4: Thank you.
Point 5: The rest of the discussion is comprehensively elaborated. The authors my consider some cutting (optional).
Response 4: In paragraph 4, our findings are compared to those of other researchers in the field. In Paragraph 5, the advantages and disadvantages of LBPP were discussed. The authors think these topics are related and important to discuss.

Reviewer 3 Report
Almutairia et al are presenting an interesting exploratoy study adressing the safety and efficacy of Lower Body Positive Pressure Treadmill Training in Individuals with Chronic Stroke. The scientific merit is unquestionable but there are minor points to be adressed: 1. Study design: define the study as an interventional study 2. This is an exploratory study based on a (very) convenient sample. The title should be changed to reflect it. Something like "The Safety and Feasibility of Lower Body Positive Pressure Treadmill Training in Individuals with Chronic Stroke: an exploratory study " would be more appropriate. 3. The implications (generalizability, external validation, bias) of some limitations ( The highly selected nature of the sample, the lack of more detailed neurological characterization of the gait and the small sample ) should be better discussed.Author Response
Response to Reviewer 3 Comments
Thank you for reviewing our manuscript on the safety and feasibility of lower body positive pressure treadmill training in individuals with chronic stroke. The authors appreciate the time and effort devoted to revising our manuscript. The responses to your comments and recommendations are provided below.
Point 1: Study design: define the study as an interventional study.
Points 2: This is an exploratory study based on a (very) convenient sample. The title should be changed to reflect it. Something like "The Safety and Feasibility of Lower Body Positive Pressure Treadmill Training in Individuals with Chronic Stroke: an exploratory study " would be more appropriate.
Response 1 & 2: Accordingly, the title has been edited.
Point 3: The implications (generalizability, external validation, bias) of some limitations ( The highly selected nature of the sample, the lack of more detailed neurological characterization of the gait and the small sample ) should be better discussed.
Response 3: Accordingly, the authors added additional sentences to the limitation section (Lines 75 – 78).

Reviewer 4 Report
Dear Authors,
thank you very much for sending the article titled:
The Safety and Feasibility of Lower Body Positive Pressure Treadmill Training in Individuals with Chronic Stroke for the review process.
The authors evaluated the safety and feasibility of LBPP as a rehabilitation tool for individuals with chronic stroke. The article seems interesting, but the authors should correct it according below comments:
- Authors wrote, that: Nevertheless, studies on the safety and feasibility of LBPP in patients with stroke are lacking. I agree with the authors, but in my opinion, firstly they should describe the problem of LBP. For example in the article titled: Prevalence and risk of spinal pain among physiotherapists in Poland, DOI 10.7717/peerj.11715, where authors determined the prevalence, symptoms of, and risk factors for spinal pain in physiotherapists, as well as analyze the correlation between these factors and the nature of the work, anthropometric features of the respondents, and the level of their physical activity. Please improve the introduction section. I suggest citing the above manuscript and writing a few sentences comparing results to "healthy" people.
- Please include ane experiment flowchart at the beginning of the Method section
- the small size of participant group resulted in large values of standard deviation
- did the authors pay attention to whether the ivolved side influenced the results of the experiment?
Author Response
Response to Reviewer 4 Comments
Thank you for reviewing our manuscript the safety and feasibility of lower body positive pressure treadmill training in individuals with chronic stroke. The authors appreciate the time and effort devoted to revising our manuscript. The responses to your comments and recommendations are provided below.
Point 1: Authors wrote, that: Nevertheless, studies on the safety and feasibility of LBPP in patients with stroke are lacking. I agree with the authors, but in my opinion, firstly they should describe the problem of LBP. For example in the article titled: Prevalence and risk of spinal pain among physiotherapists in Poland, DOI 10.7717/peerj.11715, where authors determined the prevalence, symptoms of, and risk factors for spinal pain in physiotherapists, as well as analyze the correlation between these factors and the nature of the work, anthropometric features of the respondents, and the level of their physical activity.
Point 2: Please improve the introduction section. I suggest citing the above manuscript and writing a few sentences comparing results to "healthy" people.
Response 1&2: I think the term "LBPP" is being misunderstood. The authors referred to this as "lower body positive pressure" (LBPP) in this study. Lower body positive pressure is an antigravity treadmill used for gait training.
Point 3: Please include ane experiment flowchart at the beginning of the Method section.
Response 3: The flowchart has been added.
Point 4: the small size of participant group resulted in large values of standard deviation.
Response 4: The authors acknowledged in the limitation section that the small sample size is a limitation. Unfortunately, we were unable to identify which standard deviation has a large value.
Point 5: did the authors pay attention to whether the ivolved side influenced the results of the experiment?
Response 5: The authors did not formally investigate the impact of involved and uninvolved sides on the study's primary purpose. However, the author believes that there should be no influence.

Round 2
Reviewer 1 Report
No further comments